# Assessing Integrated Hydrologic Model: From Benchmarking to Case Study in a Typical Arid and Semi-Arid Basin

**Zheng Lu** * , **Yuan He** and **Shuyan Peng**

State Key Laboratory of Earth Surface Processes and Resource Ecology, School of Natural Resources,
Faculty of Geographical Science, Beijing Normal University, Beijing 100875, China
* Correspondence: legend.lz@mail.bnu.edu.cn

**Abstract:** Groundwater-surface water interactions play a crucial role in hydrologic cycles, especially in arid and semi-arid basins. There is a growing interest in developing integrated hydrologic models to describe groundwater-surface water interactions and the associated processes. In this study, an integrated process-based hydrologic model, ParFlow, was tested and utilized to quantify the hydrologic responses, such as changes in surface runoff and surface/subsurface storage. We progressively conducted a complexity-increasing series of benchmarking cases to assess the performance of ParFlow in simulating overland flow and integrated groundwater-surface water exchange. Meanwhile, the overall performance and the computational efficiency were quantitatively assessed using modified Taylor diagrams. Based on the benchmarking cases, two case studies in the Heihe River Basin were performed for further validation and to diagnose the hydrologic responses under disturbance, named the Bajajihu (BJH) and Dayekou (DYK) cases, respectively. Both cases were 2D transects configured with in-situ measurements in the mid- and downstream of the Heihe River Basin. In the BJH case, simulated soil moisture by ParFlow was shown to be comparable with in-situ observations in general, with Pearson's correlation coefficient (R) > 0.93 and root mean square difference (RMSD) < 0.007. In the DYK case, seven scenarios driven by remote sensing and reanalysis data were utilized to study hydrological responses influenced by natural physical processes (i.e., precipitation) and groundwater exploitations (i.e., pumping) that are critical to surface and subsurface storage. Results show that subsurface storage is sensitive to groundwater exploitation before an obvious stationary point. Moreover, a correlation analysis was additionally provided demonstrating the impacts of different factors on subsurface storage timeseries. It was found that pumping influences subsurface storage remarkably, especially under short-term but large-volume pumping rates. The study is expected to provide a powerful tool and insightful guidance in understanding hydrological processes' effects in arid and semi-arid basins.

**Keywords:** groundwater; groundwater-surface water interactions; numerical modeling; benchmarking; model assessment; hydrological processes; Heihe River Basin

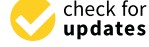



## 1. Introduction

Surface water and groundwater are essential elements in global and regional hydrologic cycles [1,2]. Surface water and groundwater, which are strongly coupled to control water and energy budgets at the catchment scale, play an important role in regulating the regional hydrologic cycle and sustainability of the local ecosystem, economy and society [3]. Quantifying the groundwater-surface water interaction is particularly vital in watershed science [4] and ecosystem function [5]. However, identifying and understanding the processes of groundwater-surface water interaction and the hydrologic responses are still difficult due to the complexity, non-linearity and heterogeneity in realistic problems [6]. In particular, hydrological processes and groundwater extraction in arid and semi-arid areas are still unclear partly due to the complex impacts of groundwater pumping and conjunctive use of groundwater and surface water for irrigation on groundwater flow

systems and exchange fluxes [7–9]. Thus, the relationship between hydrological processes and the effects of groundwater extraction in arid and semi-arid areas remains unclear.

Integrated hydrologic models are often utilized for studying the interaction between surface flow and variably saturated groundwater flow and quantitively describing hydrologic responses [10]. A wide variety of such models have been developed using different numerical algorithms and methodologies based on physical processes [11], which, however, is rather complicated yet challenging to describe such processes associated with groundwater-surface water interactions in terms of mathematical representations, discretization strategies, and computational costs [12]. The accuracy and uncertainty of integrated groundwater-surface water models principally depend on [13]: (1) the selection of governing equations (e.g., 3D Richards equation for groundwater flow and 2D approximation of the Saint-Venant equation for surface flow) [14], (2) the numerical approach (i.e., finite difference/element/volume method) chosen to solve the governing equations [15], (3) the accuracy and uncertainty of hydrometeorological data used as forcing inputs [16], (4) the discretization method (i.e., structured/semi-unstructured/unstructured mesh type) and the resolution [17], and (5) the model structural uncertainty and simplified representations of hydrological processes [15,18], and (6) modeling domain and boundary condition setups [19,20].

Therefore, it is crucial to verify the accuracy and quantify the uncertainty in these aspects for coupled groundwater-surface water modeling approaches. There have been several collaborative hydrologic model intercomparison campaigns aiming to evaluate model performance and benchmarking, such as the integrated hydrologic model intercomparison project [6,12]. The numerical experiments are usually categorized as (1) simplified controlled experiments, within which the reference data used for comparison are usually experimental data, analytical solutions or certifiably existing results [21], and (2) realistic applications, where in-situ observations or high-quality remote sensing and reanalysis products are frequently used for benchmarking and validation. However, the existing hydrologic model intercomparison researches [22–24] focuses more on qualitative evaluations (e.g., a visual comparison and empirical analysis) of model performance, which may not be sufficient to elaborate on quantitatively diagnosing (e.g., using indexes and statistical analysis) hydrologic responses of simulation results. In the current study, we intend to provide quantitative assessments of integrated hydrologic models using numerical experiments in both categories.

In the current research, several benchmarking cases and two case studies were built to investigate the following scientific questions: (1) How to evaluate the performance and computational efficiency of integrated groundwater-surface water models when estimating hydrologic behaviors? (2) How to use the information from available data on groundwater and surface water systems in order to inform the building process of conceptualizations and hydrological models in arid and semi-arid regions? (3) What are the mechanisms by which climate change and water use alter the surface and subsurface water storage in arid and semi-arid regions? We first present several classical benchmarking cases (including different scenarios as well) following previous studies [6,21,25–29]. Selected benchmarking cases include two overland flow-only cases and two integrated groundwater-surface water cases. All the benchmarking simulations are mainly performed using open-source, parallel-performance, object-oriented simulators, ParFlow (Parallel Flow) [30], which has been authenticated in simulating high-resolution coupled water and energy processes in real-world problems [31–33]. Simulated results are compared with analytical solutions (if available) and validated results from laboratory experiments and numerical simulations from selected solvers. Hydrologic response performances of ParFlow simulations are investigated using benchmarking cases from simplicity to complexity. Performances including four statistical indices are employed in the synthetical assessment. The purpose of this section is to provide a reference point for the performance of several efficient, object-oriented codes for groundwater simulations on the same platform, which involves hypothetical and real-world subsurface problems where complexity increases. The results can help design

software to effectively use resources involving consideration of an increasingly complex data layout and data access patterns.

Finally, two case studies selected in a typical arid and semi-arid region in China, the Heihe River Basin, are conducted for further validation and application. The Heihe River Basin is the second-largest inland river basin located in Northwest China [34,35]. Meteorological variables, such as air temperature and precipitation, exhibit distinct characteristics from upstream alpine regions to downstream arid regions. Additionally, it is noteworthy that the water cycle is closed in the inland rivers, which makes the Heihe River Basin an ideal study area to explore regional hydrological processes [36]. Previous studies have mainly focused on how to regulate and optimize water resources and management [37–40]. Investigations on groundwater-surface water interactions and corresponding hydrological responses during the process of overland flow generation are still in progress in the Heihe River Basin. Specifically, the midstream of the Heihe River Basin, covering oasis-desert regions, has been significantly impacted by anthropic activities, such as farm developments [41]. Thus, studying short-term hydrological responses under various disturbances is particularly interesting and important. Here we first build a test case in the Bajajihu (BJH) area (downstream of the Heihe River Basin) to validate our model in simulating hydrological responses. Simulated results are compared with soil moisture observations at different soil depths. Spatio-temporal distributions of soil moisture profiles are also analyzed. Secondly, a 2D transect is configured with in-situ measurements and driven by remote sensing and reanalysis data in the Dayekou (DYK) area (mid-stream of the Heihe River Basin). Seven scenarios are further performed to explore local hydrological responses with different rainfall rates, evaporation rates and pumping strategies.

In the following sections, the governing equations and the coupling strategy of ParFlow are briefly introduced. Then, the assessment methodology and detailed descriptions of benchmarking cases are explained, followed by two case studies in the Heihe River Basin with relevant analyses. Finally, conclusions are provided.

## 2. Methodology

In this section, the integrated hydrologic model used in the current study (ParFlow) is introduced briefly. The governing equations of overland flow and variably saturated groundwater flow are discussed in detail.

### 2.1. Integrated Hydrologic Model: ParFlow

The open-source and object-oriented simulator, ParFlow [25], which is developed to simulate integrated hydrology, is employed in this study. ParFlow is a variably saturated groundwater-surface water flow solver that considers the saturated zone, vadose zone, and surface water as an entirely hydrologic continuum based on the 3D variably-saturated Richards equation and the 2D kinematic wave approximation of the Saint-Venant equation [30].

A cell-centered finite difference approach is used in ParFlow to solve the 3D Richards equation. A globally implicit time stepping is adopted, and a Newton-Krylov method is required to solve non-linearities at every time step with multi-grid preconditioning [42]. For surface routing, ParFlow employs an upward finite volume approach and a backward Euler approach for the discretization in time of the 2D kinematic equations [43]. A free-surface overland flow matching boundary condition is used via a pressure continuity manner which enables the consistency between the kinematic wave equation (surface) and the Richards equation (subsurface) [44]. More details of ParFlow are given in [25,45], and here only a brief summary is listed in Table 1.

**Table 1.** A summary of ParFlow.

| Numerical Methods | ParFlow |
|---|---|
| Subsurface flow governing equation | Richards |
| Surface flow governing equation | Kinematic wave |
| Subsurface numerical approach | Finite difference |
| Surface numerical approach | Upward finite volume |
| Saturated-unsaturated coupling | Entire continuum |
| Subsurface-surface coupling | Free-surface boundary condition |
| Coupling strategy | Implicit |
| Discretization | Rectangular |
| Grid capacity | Structured and semi-unstructured |

### 2.2. Benchmarking Case Descriptions

As shown in Figure 1a, the complexity, heterogeneity and non-linearity of the ground-water-surface water interactions control the accuracy and uncertainty in integrated hydro-logic modeling [46]. It is then natural and essential to benchmark and validates models from simplified representations (Figure 1b, a tilted V-catchment area used for exploring hydrologic responses due to rainfall events) to beyond.

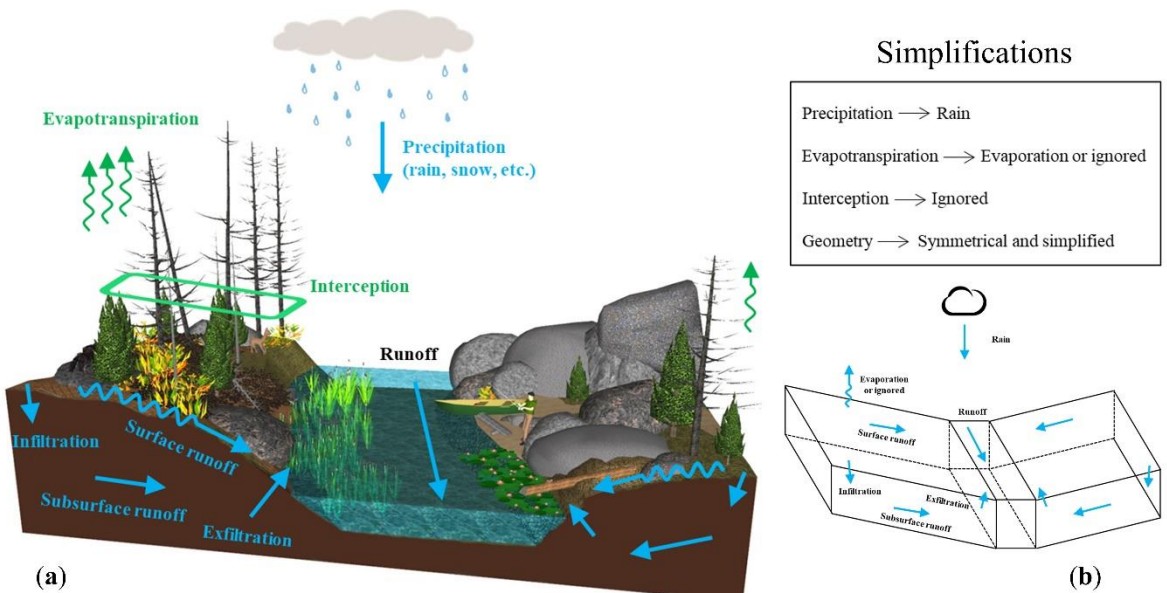

**Figure 1.** From complexity to simplicity: realistic and idealized watersheds: (**a**) conceptual model of the real-world hydrologic cycles; (**b**) an idealized tilted V-catchment model.

Four benchmarking cases in this study are summarized as shown in a schematic flow chart (Figure 2) and classified as: overland flow only and integrated groundwater-surface water. Based on the confidence in building and evaluation of these benchmarking cases, two case studies in the Heihe River Basin (BJH and DYK cases) are presented in Section 4.

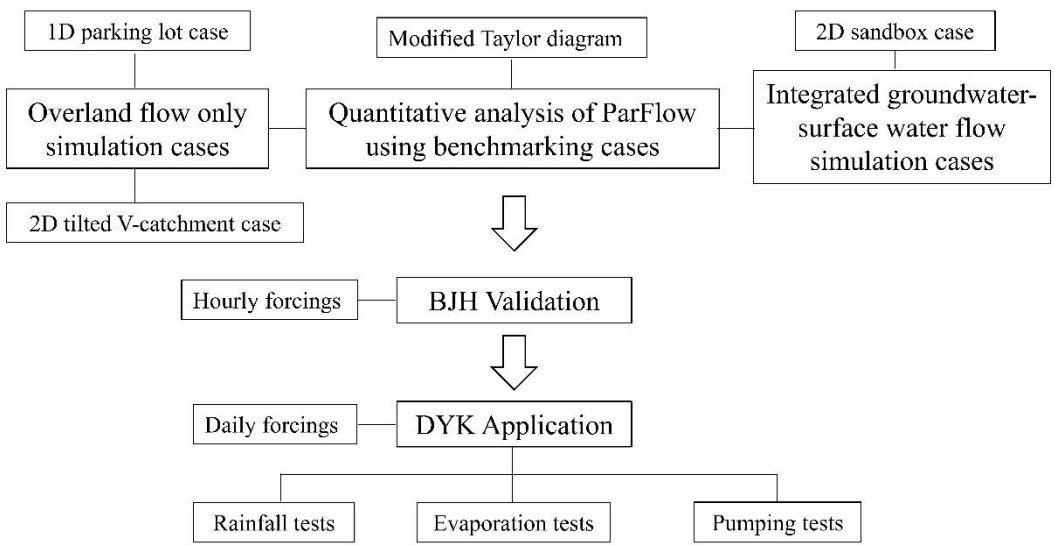

**Figure 2.** Flow chart: from benchmarking cases to the Heihe River Basin case studies.

First, the assessment methodology is presented for a comprehensive evaluation of hydrologic models. Then the benchmarking cases are described, implemented by ParFlow and compared with the analytical solution, laboratory experiment or other numerical solution (shown in Table 2). Additionally, several well-established hydrologic models with capabilities to simulate the designated benchmarking case are also included for auxiliary comparisons. In the 1D parking lot case, the analytical solution is used as the reference, with PFLOTRAN and Cast3M used as auxiliaries. Given that there is no existing analytical solution or experiment data in the 2D tilted V-catchment case, the numerical solution by Di Giammarco et al. [47], which is well documented for this case, is employed as the reference; while PFLOTRAN and WATLAC are selected as auxiliaries. As for the 2D sandbox case, the laboratory experiment is utilized while InHM and ISWGM are involved as auxiliaries.

**Table 2.** A summary of reference data used in the benchmarking cases.

| Category | Case | Reference | Auxiliary |
|---|---|---|---|
| Overland flow | Case 1: 1D parking lot | Analytical solution [26,27] | PFLOTRAN, Cast3M |
| | Case 2: 2D tilted V-catchment | Di Giammarco et al. [47,48] | PFLOTRAN, WATLAC |
| Integrated groundwater-surface water flow | Case 3: 2D sandbox | Laboratory experiment data [49] | InHM, ISWGM |

### 2.2.1. Assessment Methodology

As previously noted, assessments of integrated groundwater-surface water models and quantitative analysis of their hydrologic behaviors are important. Previous research tended to focus on a certain aspect, such as qualitative analysis of the overall performance [6,12,21,24,50], or the computational efficiency [51,52]. In this study, to implement the statistical indices, a modified Taylor diagram is used as the synthetic tool. Taylor diagrams are originally designed to graphically determine which of the given models is the most "realistic". It graphically shows the degree of agreement between the target model and the references in terms of three statistics: SD (Standard deviation), RMSD (root mean square difference), and R (Pearson's correlation coefficient).

For SD and RMSD, the value would be 0 for a perfect match or ideal model while for R, the value would be 1 for a perfect match. More information about statistical indices can be found in [53]. SD is depicted by polar coordinates. R is equal to the azimuthal angle.

The centered RMSD of the target model is proportional to the distance from the reference's coordinate on the x-axis. Here we add computational efficiency into the Taylor diagram with the size of the point. In one specific case, the shape of minimal time consuming (CPU time) is set to 1, and sizes of other simulations are multiplied by time-consuming gains. That is, the smaller size, the higher efficiency.

2.2.2. Benchmarking Case 1: 1D Parking Lot Case

Two overland flow-only benchmarking cases are involved in this section in order to evaluate the performance of simulating shallow overland flow by ParFlow. The computational domains are illustrated in Figure 3, with detailed parameters listed in Table 3. The performance of ParFlow is first tested by simulating the overland flow hydrograph associated with uniform rainfall in a simple 1D test case [26]. An effective rainfall continues for 1800 s with an intensity of $1.4 \times 10^{-5}$ m/s over a 180 m long parking lot. The slope is 0.0016, and Manning's roughness is $4.2 \times 10^{-4}$. The initial condition is a dry bed state, and a dynamic discharge rate is applied at the inlet over time.

**Table 3.** Parameters in the 1D parking lot case and 2D tilted V-catchment case.

|  |  | Unit | 1D Parking Lot | 2D V-Catchment | 2D Sandbox |
|---|---|---|---|---|---|
| Model geometry | Horizontal size | m | 180 | $1620 \times 1000$ | $1.4 \times 0.08$ |
|  | Horizontal resolution | m | 1.8/18 | 20 | 0.01 |
|  | Vertical resolution | m | 0.5 | 0.5 | 0.01 |
| Time configuration | Simulation period | s | 3600 | 10,800 | 1500 |
|  | Rain duration | s | 1800 | 5400 | 1200 |
|  | Recession duration | s | 1800 | 5400 | 300 |
|  | Time step size | s | 60 | 60 | 10 |
| Boundary conditions | Lateral and bottom | | No flow | | |
|  | Surface toe | | Overland flow | | |
|  | Overland flow | | Zero depth gradient outlet | | |
| Initial condition | Water table | m | Subsurface saturated | Subsurface saturated | 0.74 above bottom |
| Surface coefficients | Rain rate | m/s | $1.4 \times 10^{-5}$ | $3 \times 10^{-6}$ | |
|  | X direction slope | - | 0.0016 | 0 (channel) 0.05 (slope) | |
|  | Y direction slope | - | 0 | 0.02 | |
|  | Manning's roughness | | $4.2 \times 10^{-4}$ | $2.5 \times 10^{-3}$ (channel) $2.5 \times 10^{-4}$ (slope) | |
| Subsurface hydraulic coefficients | Saturated hydraulic conductivity | m/s | | | $3.5 \times 10^{-5}$ |
|  | Specific storage | $m^{-1}$ | | | $10^{-4}$ |
|  | Porosity | - | | | 0.34 |
|  | | | Van Genuchten Parameters | | |
|  | Pore-size radius ($\alpha$) | $m^{-1}$ | | | 2.4 |
|  | Pore-size distribution (n) | - | | | 5.0 |
|  | Res. vol. water content | - | | | 0.2 |
|  | Sat. vol. water content | - | | | 1.0 |

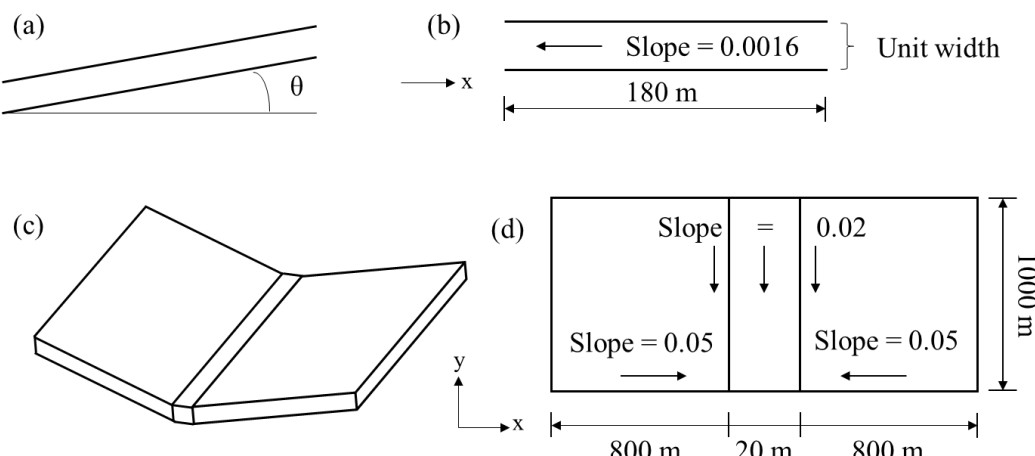

**Figure 3.** Computational domains of the 1D parking lot case and 2D V-catchment case: (**a**) Schematic description of the parking lot; (**b**) 1D view of the parking lot; (**c**) Schematic description of the V-catchment area; (**d**) 2D view of the V-catchment.

### 2.2.3. Benchmarking Case 2: 2D Tilted V-Catchment Case

The overland flow generated by a rainfall event is then simulated on a simple tilted V-catchment [47,48]. The computational domain of the tilted V-catchment is constructed by two inclined planes (800 m in length and 1000 m in width) connected by a sloping channel (20 m in width). The surface slopes are 0.05 symmetrically perpendicular to the channel for the two sloping planes and 0.02 parallel to the channel for the whole domain. The Manning's roughness values are $2.5 \times 10^{-4}$ for the sloping planes and $2.5 \times 10^{-3}$ for the channel. The precipitation includes 5400 s of rain, with a rainfall rate of $3 \times 10^{-6}$ m/min, followed by a subsequent 5400 s recession. The subsurface is assumed to be initially saturated; therefore, only overland flow activated by rainfall contributes to the outlet rate.

### 2.2.4. Benchmarking Case 3: Integrated Groundwater-Surface Water Flow

To explore the performance of ParFlow's integrated groundwater-surface water flow, two integrated benchmarking cases are involved in this section. Four scenarios are tested in the second sloping plane benchmarking cases. The geometries are demonstrated in Figure 4 with hydrogeologic properties listed in Table 3.

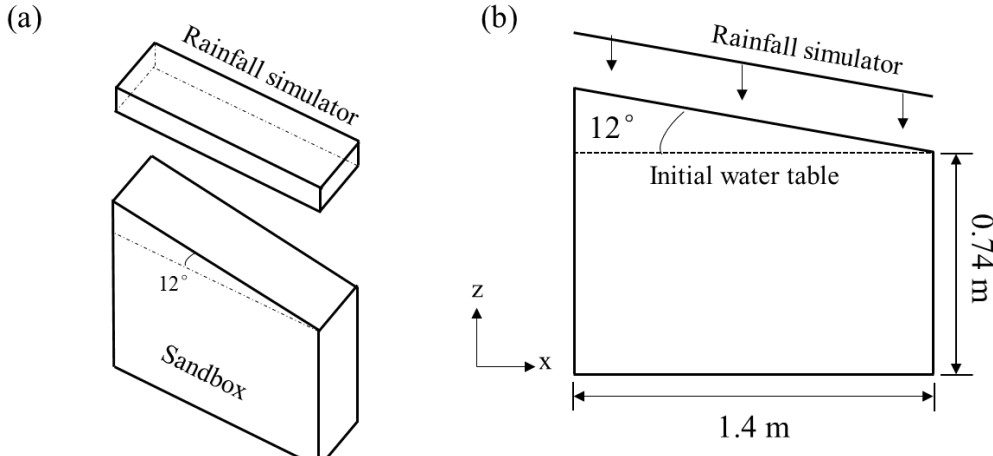

**Figure 4.** Computational domains of the 2D sandbox: (**a**) Schematic description of the sandbox; (**b**) 2D view of the sandbox.

Assessment of coupled groundwater-surface water flow was first performed by simulating a laboratory experiment conducted by Abdul and Gillham (1984) [28]. Few experimental data are available in the literature to validate integrated surface water-groundwater models, and the Abdul and Gillham experimental system is an optimal choice [49,54], which is designed to explore the rapid capillary zone responses [55].

The experimental setup consists of a Plexiglas sandbox, the dimension of which is 140 cm long, 8 cm wide and 120 cm high (shown in Figure 4a,b). Medium-to-fine sand is packed in the box forming a 12° sloping surface with its toe at a height of 74 cm from the bottom (Figure 4b). The total porosity is 0.34, and the saturated conductivity is equal to $3.5 \times 10^{-5}$ m/s. A mimetic rainfall is applied uniformly over the surface at a rate of $1.2 \times 10^{-5}$ m/s for 1200 s while the discharge volume is measured for 60 s more. The initial water table is located at the toe of the sloping surface, and the initial condition is hydrostatic.

### 2.3. Validation Case: The Bajajihu (BJH) Case

#### 2.3.1. Modeling Domain

As depicted in Figure 5a, the Bajajihu (BJH) case is a 2D transect (850 m long) selected between two hydrometeorological observatory sites in the downstream of the Heihe River Basin: the *Populus euphratica* Site in the northwest with an elevation of 874 m and the Mixed Forest Site in the southeast with an elevation of 876 m. The BJH case consists of a simple 2D hillslope covered by bare land as well as sparse *Populus euphratica* and *Tamarix*. Based on a pre-analysis of subsurface hydraulic coefficients, the effects of subsurface heterogeneity are not significant in the modeling domain. Thus, the hydraulic parameters were considered statistically homogeneous. Subsurface properties such as porosity and saturated hydraulic conductivity are considered spatially homogeneous, as shown in the Digital Soil Mapping Dataset in the Heihe River Basin [56,57] and the GLobal HYdrogeology MaPS (GLHYMPS) [58]. Hydrometeorological variables and fluxes were collected automatically from the *Populus euphratica* Site and Mixed Forest Site at different time intervals [35]. In the BJH case, evaporation, precipitation, groundwater table depths and surface soil moisture at depths of 2 cm and 4 cm are used as forcing data and initial/boundary conditions. The in-situ observation datasets are downloaded from the National Tibetan Plateau/Third Pole Environment Data Center at https://data.tpdc.ac.cn/ (accessed on 15 March 2023) supported by the Heihe Watershed Allied Telemetry Experimental Research (HiWATER) project [36].

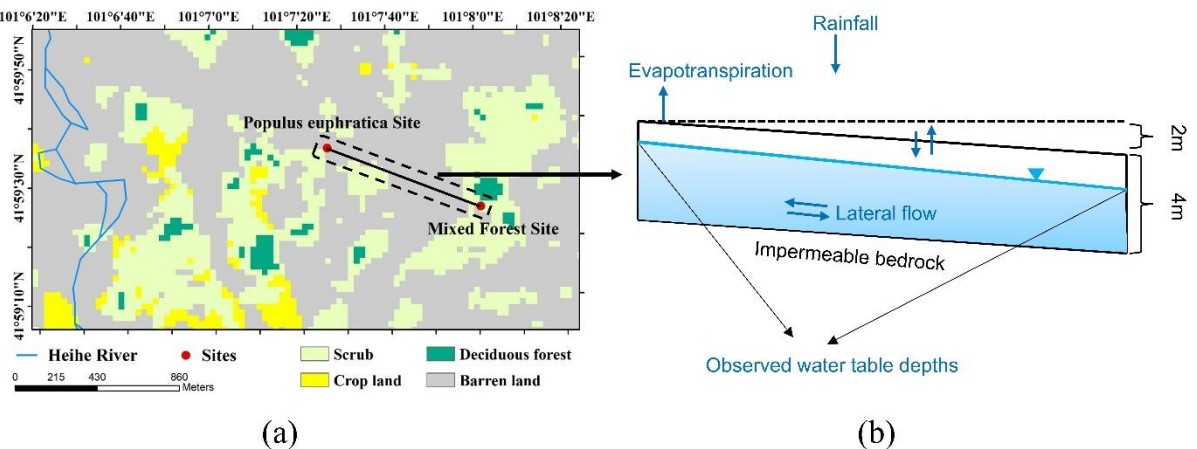

**Figure 5.** Configurations of the BJH case: (**a**) Location of the 2D transect; (**b**) Simulation setup.

#### 2.3.2. Model Setup

The 2D computational domain is shown in Figure 5b, which is 850 m in length and 4 m in height. The lateral resolution (i.e., the x direction) of the domain is 10 m, and the vertical

resolution (i.e., the z direction) is 0.01 m. Parameters such as porosity, saturated hydraulic conductivity, Manning's roughness and Van Genuchten parameters are all considered homogeneous and obtained from [59,60] (Table 4). The parameters were validated using a combination of several geologic maps (e.g., [61–63]). The front, back and bottom boundary conditions are no flow boundaries. The left and right boundary conditions are time-series observed water tables.

**Table 4.** Parameters in the BJH cases.

| | | Unit | Value |
|---|---|---|---|
| | Domain size | m | $850 \times 1 \times 4$ |
| Model geometry | Horizontal resolution | m | 10 |
| | Vertical resolution | m | 0.01 |
| | Saturated hydraulic conductivity | m/s | $1.0 \times 10^{-5}$ |
| | Porosity | | 0.22 |
| Subsurface hydraulic coefficients | Van Genuchten Parameters Pore-size radius ($\alpha$) | $m^{-1}$ | 5.0 |
| | Pore-size distribution (n) | | 1.6 |
| | Res. vol. water content | | 0.015 |
| | Sat. vol. water content | | 0.29 |
| | Evaporation rate | m/s | $4.17 \times 10^{-5}$ |
| Surface coefficients | Rain rate | m/s | Observed |
| | Slope | | $-0.0024$ |
| | Manning's roughness | | $2.5 \times 10^{-3}$ |
| Initial condition | Groundwater table depth | m | Steady-state |
| | Bottom | | No flow |
| Boundary conditions | Front and back | | No flow |
| | Left and right | | Time-series observed |

Before simulations, a series of preprocessing sensitivity was performed to identify the sensitive parameters which have huge impacts on soil moisture simulations. The results showed that the hydraulic conductivity ($K$), pore-size radius ($\alpha$) and residual water content (*Res. vol. water content*) affect the simulated soil moisture results largely. Then the calibration and testing were performed to identify the best model that is consistent with the soil moisture variations observed in the BJH domain. Initial estimates of hydraulic parameters were selected as averaged values of the modeling domains based on the two databases mentioned above. Additionally, the K, $\alpha$ and Res. vol. water content were further manually calibrated by comparing simulated and observed values of soil moisture during 31 August–2 September 2015. Finally, the BJH case was simulated using ParFlow on 3 September 2015. The time step of the simulation is set to 600-s, the same as the interval of precipitation observation. The groundwater table depth is originally at an interval of 1800-s and interpolated linearly. Input data of the precipitation and groundwater table depths on 3 September 2015 are shown in Figure 6. Meanwhile, the evapotranspiration rate, including bare soil evaporation and vegetation transpiration, is daily averaged. Before simulations, sensitivity analysis and preprocessing calibration were performed to adjust several parameters to obtain better simulation results.

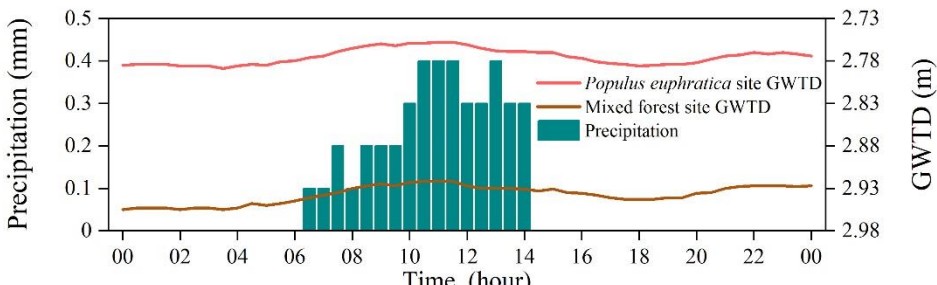

**Figure 6.** Time-series of precipitation and groundwater table depth (GWTD) on 3 September 2015 for the BJH case.

### 2.4. Application Case: The Dayekou (DYK) Case

#### 2.4.1. Modeling Domain

The Dayekou (DYK) case represents the middle reaches of the Heihe River Basin. The elevation of the selected 2D transect increases from 2300 m in the northeast to 2500 m in the southwest. As shown in Figure 7, a horizontal 800 m long section (first introduced in [64]) is defined in the DYK area. The DYK section involves a simple 2D hillslope. Local landscapes include an alpine meadow, dry desert, cropland (barley) and the DYK irrigation ditch. The primary mountain overland flow and infiltration are generated by precipitation in the southwest. The surface water is flown from the northeast following the terrain and discharged with the DYK irrigation ditch. At the same time, bare soil evaporation and vegetation transpiration influence the local water cycle. During the crop-growing seasons, a pumping well on the barley farmland has a remarkable impact on the local runoff and subsurface storage. As for the subsurface, the shallow area is obviously stratified, and here we simplify it with two layers [65].

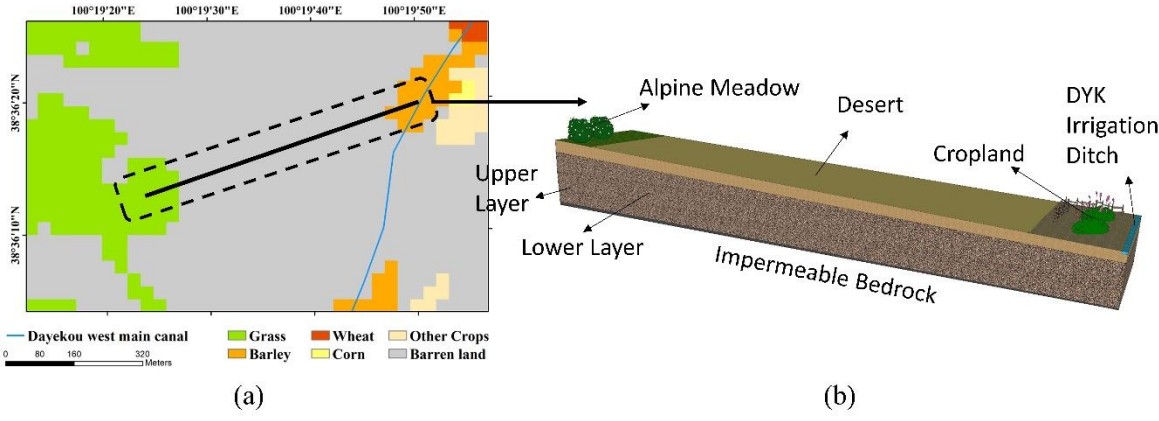

(a)                                                                (b)

**Figure 7.** Conceptual model of the DYK case: (**a**) Location; (**b**) Landcover, hydrological and geological conditions.

#### 2.4.2. Model Setup

The 2D domain selected to perform this application is simplified from actual geology and sketched in Figure 8, marked with primary model inputs (length and elevation, potential recharge parameters, surface and subsurface conditions and well location). Here seven scenarios are utilized to investigate regionally hydrological processes with different rainfall rates, evaporation rates and pumping strategies. The domain dimensions are 800 m in length and 7 m in height. Impermeable bedrock is assumed at the bottom of the section, and no flow boundary conditions are allocated to left and right boundaries. A pre-analysis of subsurface hydraulic coefficients showed that subsurface heterogeneity is not significant in the DYK modeling domain. Thus, the hydraulic parameters were considered statistically homogeneous in the vadose zone and saturated zone. For simplifications, homogeneous

surface hydrometeorological elements (i.e., rainfall, evaporation) are used. Subsurface properties (i.e., porosity, saturated hydraulic conductivity and Van Genuchten parameters) are considered horizontally homogeneous and vertically divided into two layers because previous studies showed coefficients with patches distribution are capable of addressing regional hydrology characteristics [66]. A pumping well is located 100 m from the right boundary for the usage of irrigation of the barley farmland.

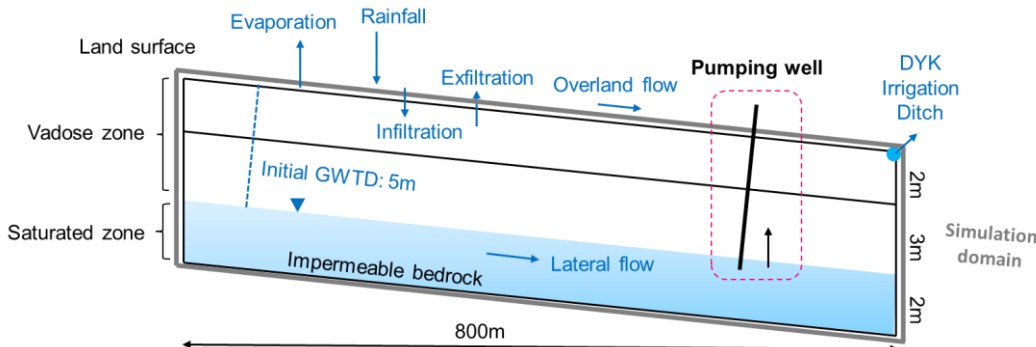

**Figure 8.** Illustration of simulation configurations of the DYK case: 2D view and basic components of the section of the DYK case.

The lateral resolution (i.e., the x direction) of the domain is 10 m and the vertical resolution (i.e., the z direction) is 0.1 m. It extends 7 m below the surface with two layers (2 m on top and 5 m below), and each layer is composed of homogeneous soils. Porosity, saturated hydraulic conductivity, Manning's roughness and Van Genuchten parameters are extracted from [59,60]. The initial groundwater table depth is set to 5 m due to several long-term local surveyors. The lateral and bottom boundary conditions are no flow boundaries. Before the simulations, several boundary condition tests were performed, and the results showed that the left and right boundaries set as no-flow conditions lead to the highest accuracy. The parameters of the DYK transect are shown in Table 5.

**Table 5.** Parameters in the DYK cases.

| | | Unit | Shallow Layer | Deep Layer |
|---|---|---|---|---|
| Model geometry | Layer depth | m | 2 | 5 |
| Subsurface hydraulic coefficients | Saturated hydraulic conductivity | m/s | $3 \times 10^{-6}$ | $8.5 \times 10^{-6}$ |
| | Porosity | | 0.4589 | 0.4102 |
| | Van Genuchten Parameters | | | |
| | Pore-size radius ($\alpha$) | $m^{-1}$ | 1.03 | 2.3 |
| | Pore-size distribution (n) | | 1.174 | 1.254 |
| | Res. vol. water content | | 0.02 | 0.04 |
| | Sat. vol. water content | | 0.437 | 0.349 |
| Surface coefficients | Slope | | 0.25 | |
| | Manning's roughness | | $2.5 \times 10^{-3}$ | |
| Initial condition | Groundwater table depth | m | 5 | |
| Boundary conditions | Lateral and bottom | | No flow | |
| | Surface toe | | Overland flow | |
| | Overland flow | | Zero depth gradient outlet | |

### 2.4.3. Forcing Data

Daily accumulated precipitation and daily evaporation data used in this case were obtained from EC-EARTH-Heihe [67] and GLEAM [68]. However, the scales of remote sensing and reanalysis products are different from the DYK section spatially. Meanwhile, daily accumulated precipitation and daily evaporation data may not reflect fluctuant transient

dynamics temporally. Fixing these gaps and obtaining optimally spatiotemporal forcing data in the DYK test case is difficult yet critical. According to local climatic conditions, we assume that a baseline situation with precipitation takes place in one-fifth of one day. For evaporation, no variation is presumed in one day thus making the transient evaporation equal to daily values. As shown in Figure 9, both daily accumulated precipitation and daily evaporation were extracted for a 5-year growing season (Jul.-Nov.) timeseries from 2013 to 2017. The rainfall rate used in the DYK baseline case, therefore, is set with a value of $6.67 \times 10^{-5}$ m/s, while the evaporation rate is 1.8 mm/day. The pumping volume data is abstracted from Monthly irrigation datasets (for both surface water and groundwater) with 30-sec spatial resolution over the Heihe River Basin [69]. All the datasets mentioned above are available at http://data.tpdc.ac.cn/en/ for downloading.

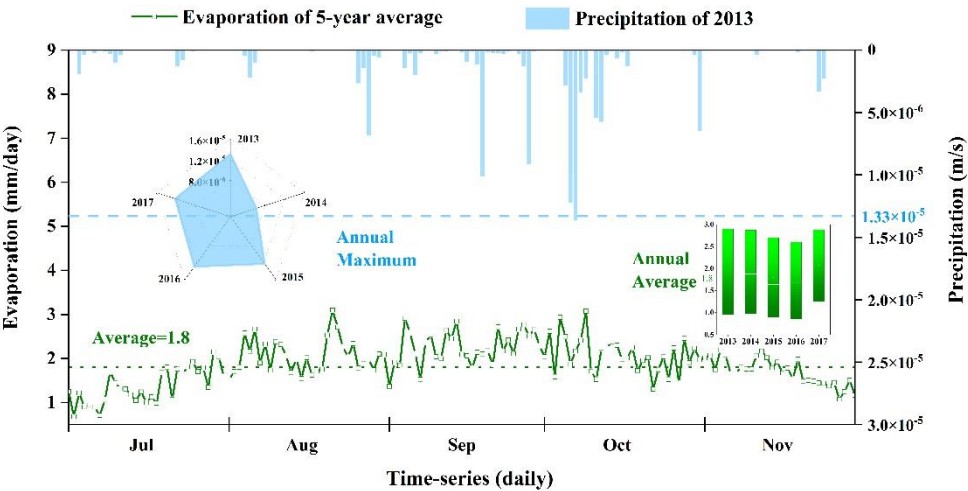

**Figure 9.** Averaged precipitation and evaporation during the period of vegetation growth season over the DYK case domain.

### 2.4.4. Scenarios

Seven different scenarios were set up and inter-compared to diagnose hydrology responses and evaluate the performance of the model. A summary of the scenarios is shown in Table 6. The baseline scenario representing normal situation of the DYK test case is first given and used as a basic reference for the following scenarios. In S1 and S2, subsurface storage changes are investigated under different evaporation situations. In S3 and S4, we focus on the influences of rainfall rate on surface/subsurface storage volumes. Finally, in S5 and S6, the effects of two irrigation groundwater scenarios on the 2D transect are explored, which ultimately aims at facilitating localized agricultural use of barley irrigations.

**Table 6.** A summary of scenarios in the groundwater-surface water modeling in the DYK case.

| Scenarios | Rainfall Rate | Evaporation Rate | Pumping Rate | Pumping Period |
|---|---|---|---|---|
| Baseline | $6.67 \times 10^{-5}$ m/s | 1.8 mm/day | 0 | - |
| 1 | $6.67 \times 10^{-5}$ m/s | 0 | 0 | - |
| 2 | $6.67 \times 10^{-5}$ m/s | 9 mm/day | 0 | - |
| 3 | $2.5 \times 10^{-5}$ m/s | 1.8 mm/day | 0 | - |
| 4 | $1.0 \times 10^{-4}$ m/s | 1.8 mm/day | 0 | - |
| 5 | $6.67 \times 10^{-5}$ m/s | 1.8 mm/day | $3.33 \times 10^{-4}$ m³/s | 0–3600 s |
| 6 | $6.67 \times 10^{-5}$ m/s | 1.8 mm/day | $1.0 \times 10^{-3}$ m³/s | 2400–3600 s |

## 3. Results and Discussion

### 3.1. Benchmarking Case 1: 1D Parking Lot Case

The results (dx = 1.8 m) of the 1D parking lot case for ParFlow are plotted as a function of time in Figure 10a using the analytical solution as reference (5% error added). PFLOTRAN and Cast3M (obtained from [29]) simulation results are used for cross-comparison as well. In general, three hydrologic models show good serial correlations, and all agree with the analytical solution, showing similar outflow rates throughout the hydrograph profile. Specifically, near the peak area, PFLOTRAN is the closest to the analytical solution. Cast3M rises relatively slower but reaches the peak as PFLOTRAN at 30 min. ParFlow adjoins the analytical solution well at the beginning of a rainfall event but shows a slow response to the rainfall. This discrepancy may be partly due to different approximation approaches of the Saint-Venant equations in the solvers. At the drainage segment, the performance of the three is all agreed with each other in general.

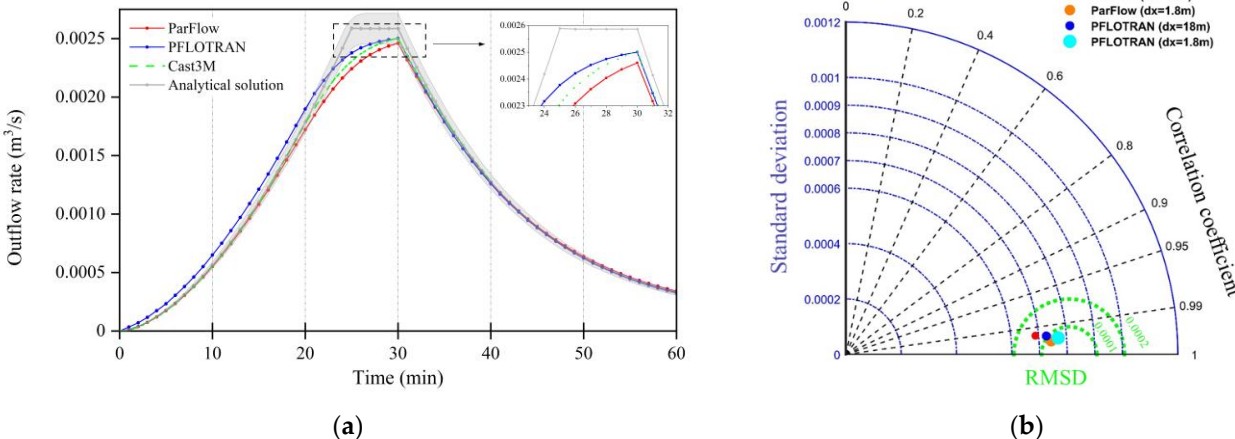

**Figure 10.** Outflow response comparison among ParFlow, PFLOTRAN, Cast3M and analytical solution for 1D parking lot case: (**a**) Time-series comparison plots. 5% error of analytical solution is drawn in gray as well; (**b**) Taylor diagram for ParFlow and PFLOTRAN, including SD, R, RMSD and normalized computational efficiency (the smaller size, the higher efficiency).

Furthermore, ParFlow and PFLOTRAN are compared under two spatial resolutions (dx = 1.8 m and 18 m). The integrated surface/subsurface solver of PFLOTRAN is mainly different from ParFlow in two aspects: (1) the governing equations for the surface flow of PFLOTRAN are the 2D diffusive wave approximation of the Saint-Venant equation; (2) the discretization algorithm of PFLOTRAN is the finite volume method. Statistical indicators for performance evaluation are depicted in Figure 10b in the form of the Taylor diagram. Compared with PFLOTRAN, ParFlow simulations have smaller SDs but larger RMSDs. From the point distribution in the Taylor diagram, it can be seen that discretization is the dominant influence factor. Thus the influence of discretization strategies is considered greater than solver differences between ParFlow and PFLOTRAN, which provide evidence supporting similar results in [21].

### 3.2. Benchmarking Case 2: 2D Tilted V-Catchment Case

Results of ParFlow along with PFLOTRAN and WATLAC are presented to validate outflow response behaviors (Figure 11a). Here WATLAC is selected because it is a distributed hydrological model, which is different from ParFlow and PFLOTRAN. The results between the three models are disparate from rainfall to recession periods. PFLOTRAN produces a very fast rise, such as the same situation in the 1D parking lot case, which may be related to the use of a bed slope instead of a water elevation slope for calculating overland flow. Performances of WATLAC and ParFlow are similar at the drainage segment but differ with the climbing rate at the rainfall segment. Near the peak area, results simu-

lated by WATLAC exhibit a small delay after 90 min probably because of its loose coupling strategy of groundwater-surface water discharge. ParFlow and PFLOTRAN both show sensitivities to the cessation of rainfall, and the recession rate predicted by ParFlow is closer to the results from Di Giammarco et al. [47]. The influence of missing the diffusive term in the kinematic wave approximation of the Saint-Venant equation does not hamper the prediction of surface water discharge using ParFlow.

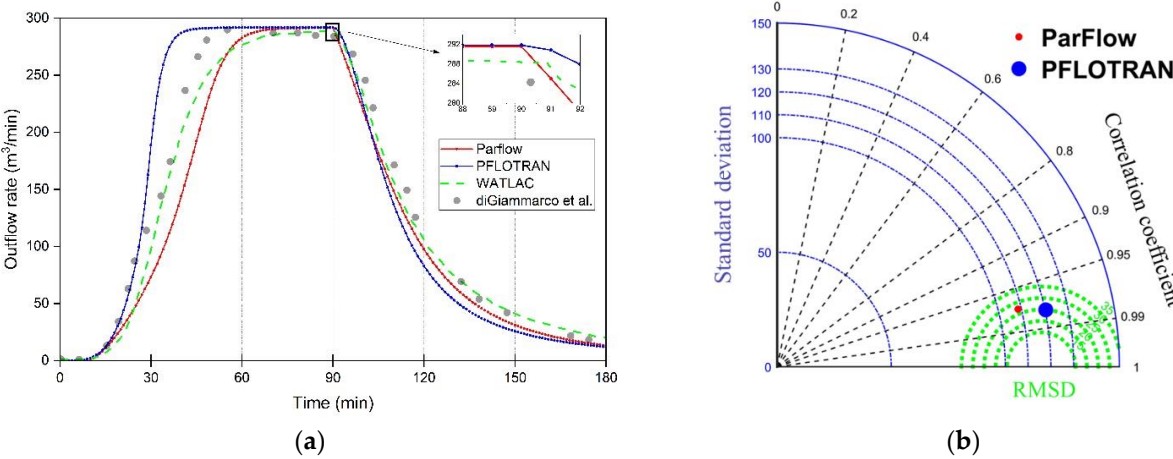

(**a**)  (**b**)

**Figure 11.** Outflow response comparison among ParFlow, PFLOTRAN, WATLAC and Di Giammarco et al. [47,48] for 2D tilted V-catchment case: (**a**) Time-series comparison plots. 5% error of analytical solution is drawn in gray as well; (**b**) Taylor diagram for ParFlow and PFLOTRAN, including SD, R, RMSD and normalized computational efficiency (the smaller size, the higher efficiency).

As shown in Figure 11b, the R and RMSD values of ParFlow and PFLOTRAN are close, while the SD value of ParFlow is lower than PFLOTRAN. This indicates that the simulation dispersion of ParFlow is lower, which means simulating hydrograph in ParFlow is more stable. Moreover, the computational efficiency of ParFlow is higher. Overall, considering the discussion above, the surface flow simulation of ParFlow agrees well with existing simulation results. ParFlow is capable of depicting hydrographs with different complexities.

*3.3. Benchmarking Case 3: Integrated Groundwater-Surface Water Flow*

Figure 12a shows the results-outflow responses of ParFlow, InHM (obtained from [12]) and ISWGM (obtained from [54]), as well as laboratory measurement data [28]. All the simulation results reach the peak of the hydrograph and drain to recession faster than experiment measurements. This "hysteresis" phenomenon was also found in [25,49,54], which can be accounted for the deficiency in dealing with air phase compression in these models.

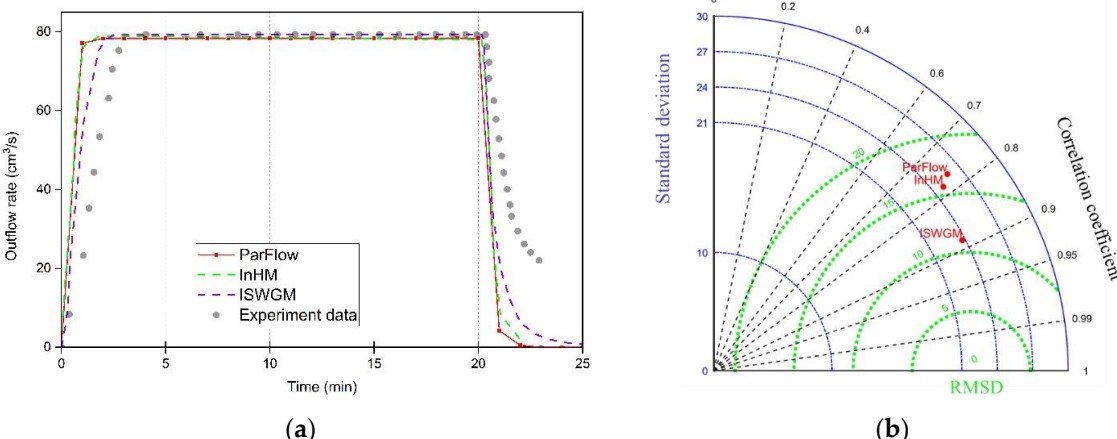

**Figure 12.** Outflow response comparison among ParFlow, InHM, ISWGM and experiment for the 2D sandbox case: (**a**) Time-series comparison plots; (**b**) Taylor diagram for ParFlow, InHM and ISWGM, including SD, R and RMSD.

Performances of ParFlow, InHM and ISWGM compared with laboratory experiments as reference are summarized and shown in Figure 12b. ISWGM performs the best in this case with the highest R, smallest RMSD and SD. Though ParFlow and InHM have larger RMSDs and SDs, they both have a strong correlation relationship (R > 0.7) with laboratory experiment data.

*3.4. Benchmarking Assessment Summary*

A series of numerical experiments using ParFlow were conducted, the results of which were compared with reference data in the four benchmarking cases above. Outflow, surface runoff and subsurface storage hydrographs and upslope distance dynamics were analyzed for examining the overland flow and groundwater-surface water flow simulations of ParFlow. Assessments were accomplished in order to identify the overall performance of model simulations by using a comprehensive Taylor diagram.

In summary, runoff processes simulated by ParFlow are consistent with an analytical solution or results from existing models when solving overland flow-only cases. Meanwhile, ParFlow has a higher stand-alone operation efficiency than PFLOTRAN for computing surface runoff. As for the integrated groundwater-surface water flow simulation cases, ParFlow performs well in terms of simple infiltration excess, saturation excess and heterogeneous slab scenarios by capturing surface runoff dynamics. Meanwhile, the subsurface variability and fluctuation of water table changes are shown in the ParFlow simulation results in the return flow scenario. All these demonstrate ParFlow's capability of quantifying the integrated hydrologic cycles with high efficiency.

## 4. Case Studies in the Heihe River Basin

*4.1. Results and Analyses of Validation Case: The Bajajihu (BJH) Case*

Surface soil moisture simulated by ParFlow is compared with in-situ observations at depths of 2 cm and 4 cm. As shown in Figure 13, simulated soil moisture agrees with in-situ observations in general, with R of 0.94 and 0.93 for 2 cm and 4 cm respectively. Besides, ParFlow simulations exhibit very low RMSD compared with in-situ values. There is a great consistency between ParFlow simulations and in-situ observations before and after the rainfall event. However, the rates and magnitudes of increase between ParFlow simulations and in-situ observations deviate during the rain. Shortly after the beginning of the rainfall, ParFlow soil moisture simulations rise quickly to the peak. Meanwhile, in-situ observations show a relatively slow and durable increasing rate. We believe that, to a large extent, the uncertainty in soil parameters leads to this issue.

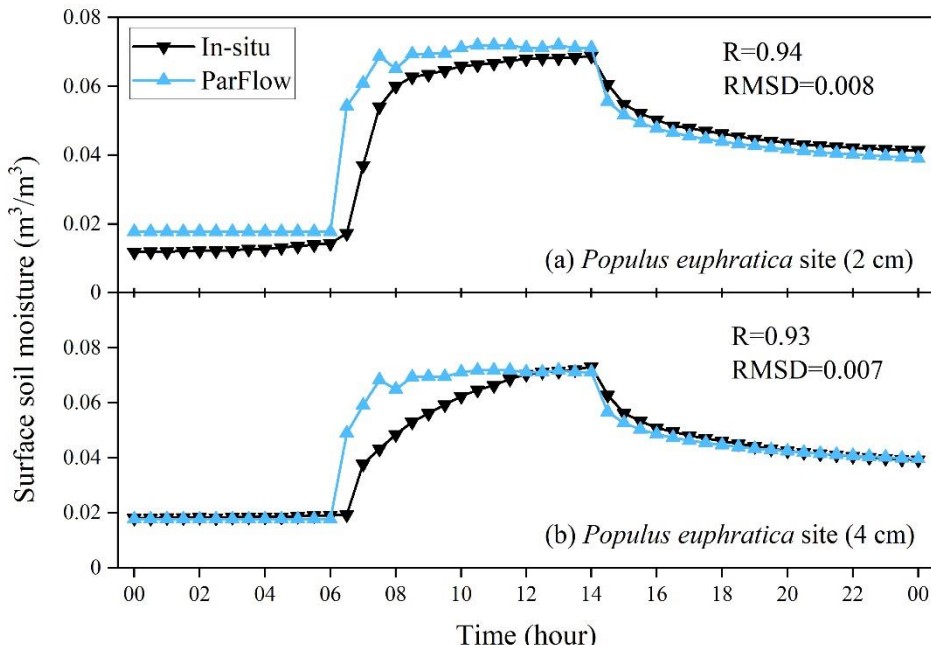

**Figure 13.** Surface soil moisture time-series comparison between ParFlow simulation and observation at the *Populus euphratica* site on 3 September 2015 for the BJH case: (**a**) 2 cm; (**b**) 4 cm.

In addition, vertical saturation distributions at four different horizontal positions ($x$ = 1 m, 28 m, 56 m, 85 m) are also drawn in Figure 14 at an interval of 1-h (same as the simulation timestep). Colors are set to represent instantaneous moments from 00:00 (red) to 24:00 (purple). From the variations of vertical saturation hydrographs, we can identify vertical saturation responses controlled by the combined impact of the rainfall rate, evaporation, water table variation and terrain. Before the rainfall event, patterns of all profiles that responded to the evaporation are almost the same. The degree of saturation during the rain period is regulated by the infiltration rate. After the rainfall event (from 17:00 to 24:00, purple lines), distributions of saturation profiles begin to be stabilized, which is depicted by the soil moisture distributions as well. As shown in the vertical saturation hydrograph at $x$ = 1 m, infiltration cause saturation from the top to groundwater table depth. Soon after the rainfall event, unsaturation occurs at the land surface due to the evaporation and distributions of saturation profiles begin to be stabilized. Saturation profiles at $x$ = 28 m and $x$ = 56 m shares a similar pattern but still can be distinguished at the beginning of the cessation period. This phenomenon is caused by the slope impact, which controls the lateral flow. However, these two sets of saturation profiles show different patterns to saturation profiles on the boundaries ($x$ = 1 m and $x$ = 85 m) due to the configuration of variable water tables. At the $x$ = 28 m position, the evolution of saturation profiles is nearly the same as that at $x$ = 1 m, except for periods of the start of the day and drainage. We believe this is probably due to the combination of water table variation and terrain impacts.

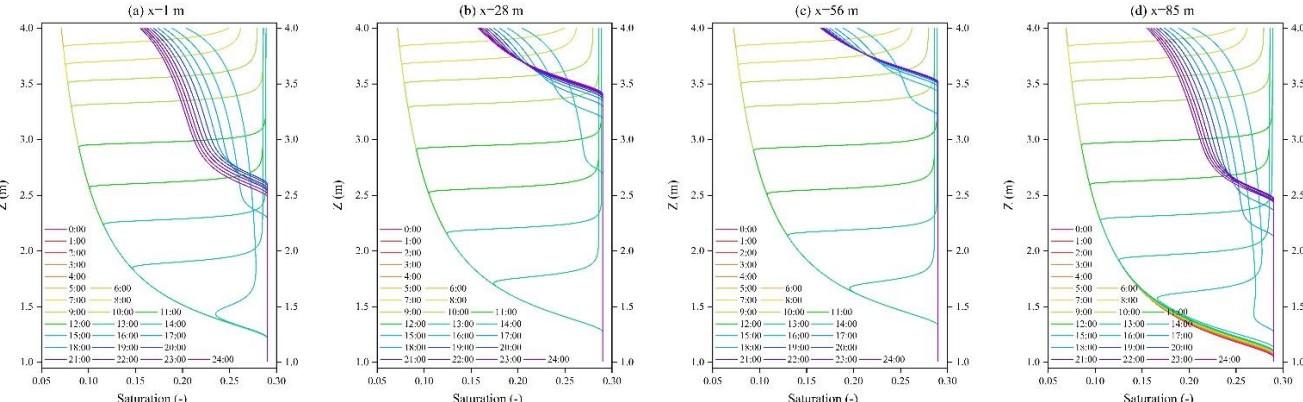

**Figure 14.** Vertical saturation distributions of at different horizontal positions: (**a**) x = 1 m; (**b**) x = 28 m; (**c**) x = 56 m; (**d**) x = 85 m for the BJH case.

*4.2. Results and Analyses of Application Case: The Dayekou (DYK) Case*

Surface/subsurface storage volume changes and discrepancies between scenarios are analyzed. Moreover, a correlation intercomparison is given. The subsurface storage results of the baseline scenario, S1 and S2 of the DYK case, are shown in Figure 15a. While subsurface storage volumes of all three increased at first nearly at the same rate, differences between S2 (representing high evaporation) and S1 (no evaporation) and baseline scenario (low evaporation) appear at 46 min. After rainfall events, subsurface storage volumes of S2 decline. On the other hand, the disparity between the baseline scenario and S1 is not distinct. The influence of 5 times evaporation rate on the subsurface storage of the DYK section is momentous, which leads to the continuous declination of the subsurface storage volume. Regional fierce drought occurred in 2014 [70,71], which can be seen as well from Figure 15 according to annual accumulated precipitation of the EC-EARTH-Heihe products. Rare rainfall and high evapotranspiration in the midstream of the Heihe River Basin may imply a potential risk of ecological unbalance [72]. Combined with the scenario analysis presented here, we predict that subsurface storage may face a big loss because of high evaporation in 2014.

Here S3 and S4 demonstrate both infiltration excess and saturation excess that generate surface runoffs in the DYK soil-vegetation section. To analyze the impacts of precipitation rate, volume dynamics of surface and subsurface storage of S3 and S4 are drawn in Figure 15b,c, as well as the baseline scenario. The difference between the baseline scenario (representing normal precipitation), S3 (weak precipitation) and S4 (strong precipitation) is greater in surface storage but not significant in subsurface storage. It shows the vital role of precipitation in surface ponding partly due to soil hydraulic properties.

Anthropogenic groundwater exploitation is remarkable (decreased the water table by about 2 m) in the midstream of the Heihe River Basin during the last decade [69]. Water cycles and soil moisture distribution are altered, showing the increasing importance to manage groundwater withdrawal associated with surface water use. Establishing the water resource management control unit is an optimal way for the reason a decentralized rural agriculture pattern is occupied in the mid-stream of the Heihe River Basin [73–75]. S5, S6 and baseline scenarios are depicted in Figure 15d to explore the influence of pumping on the subsurface storage of the DYK section. S5 causes an approximate 6 m$^3$ loss in the subsurface storage of the entire section. From the comparison between the baseline scenario and S6, we see an obvious stationary point, which means the subsurface water volume in the DYK domain is very sensitive to pumping. To some extent, this case demonstrates ParFlow's capability to simulate groundwater exploitation and capture its effects. More detailed studies can be found in [76]. In our future research, we will investigate the impacts of groundwater exploitation in selected study sites forced by high spatiotemporal resolution remote sensing products.

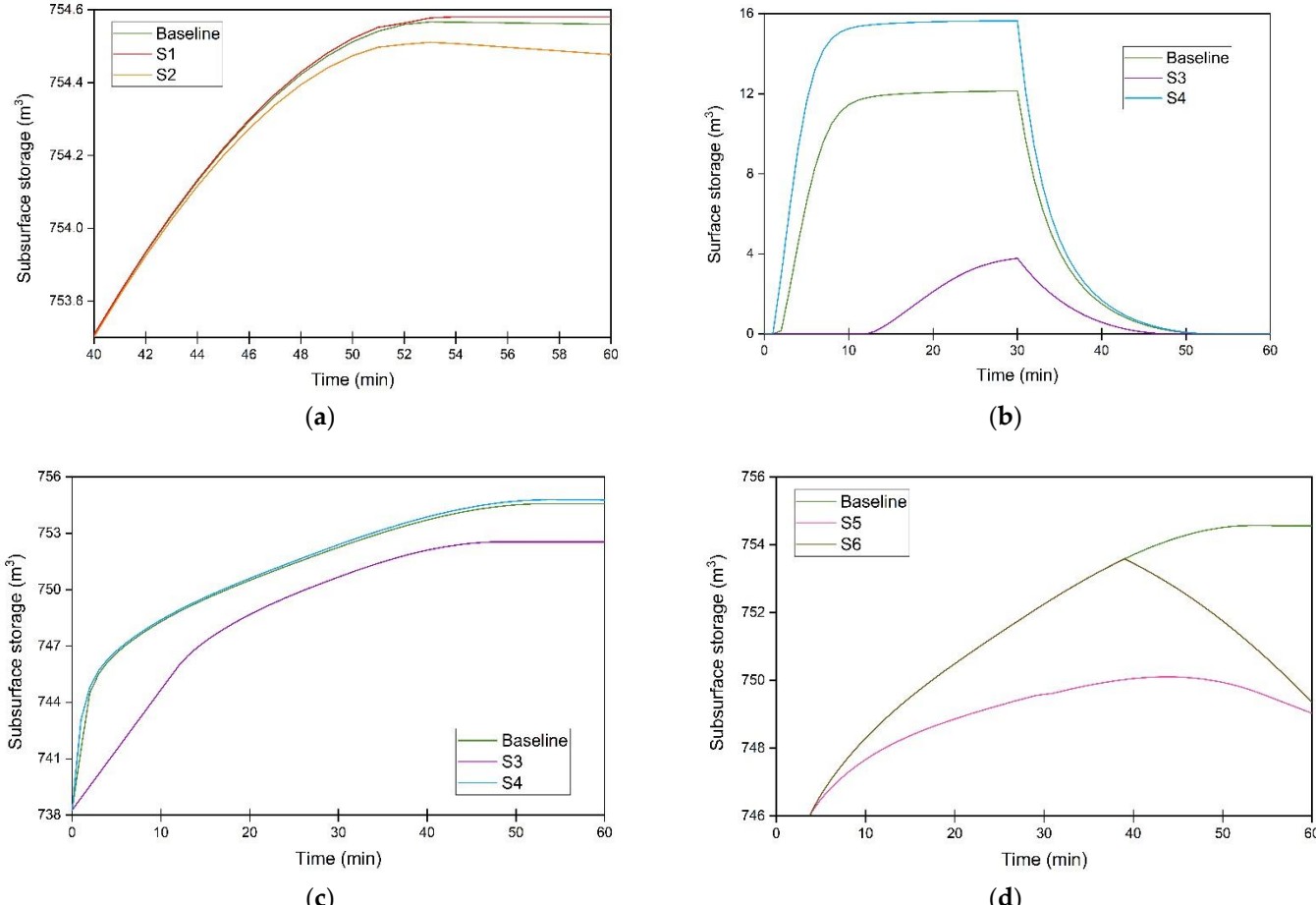

**Figure 15.** Comparison between storage time-series for the DYK case: (**a**) subsurface storage time-series of baseline, S1 and S2; (**b**) surface storage time-series of baseline, S3 and S4; (**c**) subsurface storage time-series of baseline, S3 and S4; (**d**) subsurface storage time-series of baseline, S5 and S6.

One of the challenges of modeling integrated groundwater-surface water flow is the accuracy and uncertainty of meteorological forcing data. In this section, we used high-precision remote sensing and reanalysis products as inputs for the model and generated seven scenarios. Correlation between scenarios was depicted to explore the impacts of different elements on subsurface storage timeseries. However, there are no uniform criteria to quantify the effects of regional precipitation and evaporation changes and single-point pumping on surface and subsurface storage. Here we still provide a promising approach to diagnose influences between precipitation and evaporation changes with single-point pumping influences.

## 5. Conclusions

The primary contribution of this work is to quantitatively asses an integrated groundwater-surface water model, ParFlow, using benchmarking cases, as well as case studies in the Heihe River Basin. Classical benchmarking cases were organized and tested in order, and ParFlow's hydrologic performances were assessed in a synthesis way. Benchmarking studies focused on comparing and assessing the differences in simulation outputs of different models (e.g., [6,12,29,54]). Here our study first used the modified Taylor diagram as the synthetic tool to build a qualitative analysis of the overall performance in benchmarks. Subsequently, based on the confidence in building and evaluating these benchmarking cases, two case studies in the Heihe River Basin were presented. ParFlow was verified and evaluated in a realistic but simplified case representing the downstream of the Heihe River Basin—the BJH. Finally driven by remote sensing and reanalysis products, ParFlow was

utilized in a more complicated site—the DYK representing the mid-stream of the Heihe River Basin.

We draw some specific highlights of this study:

1.  Benchmarking cases are summarized and categorized progressively from simplicity to complexity. The benchmarking cases cover a nearly full range of typical benchmarking cases to diagnose hydrologic responses, which enables subsequent users to build confidence. In each case, selected references are involved in validating results simulated by ParFlow. Generally, ParFlow can simulate the hydrological response for overland flow and integrated groundwater-surface water systems.

2.  An overall performance assessment of corresponding hydrologic signals is explored and applied using modified Taylor diagrams, which demonstrates ParFlow's capability of quantifying integrated hydrologic cycles with high efficiency. It can improve understanding of evaluation methodology and enhance the quantitative analysis of groundwater-surface water modeling.

3.  A 2D transect is configured in the BJH, with in-situ observations as inputs. Results simulated by ParFlow are validated using in-situ soil moisture observations and analyzed using soil moisture profiles and vertical saturation distributions, which in total demonstrates the capability of ParFlow in describing hydrological responses in the HRB. It is noted that we are extending the current research towards a comprehensive assessment of the integrated hydrological model in simulating 3D groundwater-surface water interactions with sensitivity analysis in the HRB.

4.  More complex conceptualization is configured in the DYK domain, with primary model geometry, and several inputs are from in-situ measurements. Meanwhile, the simulations are driven by remote sensing and reanalysis products. Seven scenarios are utilized to investigate hydrological responses influenced by natural processes and groundwater exploitations. Surface runoff and subsurface storage volume show different sensitivities to perturbations such as precipitation, evaporation and groundwater exploitation. Moreover, groundwater exploitation is proved to be more influential than natural precipitation and evaporation anomalies using a correlation coefficients heat map.

A hydrologic model of conjunctive groundwater-surface water flow, ParFlow, was used as a representative physically-based, spatially-distributed model here to describe the nonlinear surface water-groundwater interactions and the hydrologic responses such as propagation of surface runoff and dynamics of subsurface storage. From simplicity to complexity and from ideality to the real-world, all the cases were simulated and evaluated under a variety of configurations and parameter combinations to detect the impact of discretization, heterogeneity and other factors on the dynamics of hydrologic responses.

However, currently, there is still a number of defects and uncertainties in hydrological modeling (e.g., enormous discrepancies in hydrograph profiles and response signals depending on model discretization or resolution). Though realistic but simplified data was used, spatial heterogeneity and complex topography are not involved, which may lead to distortions in hydrological processes and correspond to different hydrological models' outputs. It is noted that the current study only employed simplified datasets and configurations, which may lead to distortions in hydrological processes and lead to biased hydrological simulation outputs. To further analyze the implications of these simplifications and the possible impacts of spatial heterogeneity and complex topography on the coupled groundwater–land surface model simulations at a larger (basin) scale, an intercomparison of integrated 3D groundwater-surface water models involving ParFlow-CLM at larger-(e.g., basin) scale is currently ongoing.

**Author Contributions:** Z.L.: Data curation; Methodology; Formal analysis; Validation; Visualization; Writing—original draft, review and editing; Funding acquisition; Supervision. Y.H.: Data curation. S.P.: Data curation. All authors have read and agreed to the published version of the manuscript.

**Funding:** This research was funded by the Strategic Priority Research Program of the Chinese Academy of Sciences (Grant No. XDA20100104) and the National Natural Science Foundation of China (Grant No. 41877183).

**Data Availability Statement:** The HRB data used in this study is available at https://data.tpdc.ac.cn/ (accessed on 15 March 2023).

**Acknowledgments:** The authors thank Chen Yang and Jin Liu for their technical supports.

**Conflicts of Interest:** The authors declare no conflict of interest.

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
