# Peer review of "Assessing Integrated Hydrologic Model: From Benchmarking to Case Study in a Typical Arid and Semi-Arid Basin"

_land, doi:10.3390/land12030697_

Round 1

Reviewer 1 Report

1. There are too many abbreviations, the reader is sometimes lost.

2. Do not use units for the roughness! Roughness is the parameter of the used surface and it is dimensionless! Manning´s roughness was derived from the Manning´s equation which is an empirical one! Normal Manning´s roughness coefficient n values are e.g. for natural streams - clean and straight n = 0,03, for major rivers n= 0,035, for ovegrown channels n= 0,05 or more, but these values are all dimensionless! (Tab.4, Tab.5, etc)

3. Fig.6 - Use groundwater table depth instead of water table depth (it is more times in the text, as well).

4. Fig.8 - Use Initial GWTD instead of Initial WTD.

5. I would appreciate your using of SI units (second instead of min.)

6. Line 341 - Evaporation (rainfall) rate (Table 6) is better (understandable) expressed in mm/day.

7. Fig.14 - There are too many vertical saturation distributions (every 30-minute interval) presented in the figure. Sometimes less is more, results will be clearer and more representative.

Author Response

We thank Reviewer 1 for his/her thorough review, positive feedbacks, and constructive suggestions. All the comments are carefully addressed, and point-by-point responses are listed in the attachment. All responses can be found in the Word file.

Reviewer 2 Report

The study presented an in-depth study of the hydrological responses in arid and semi-arid basins using an integrated hydrologic model, ParFlow. The case studies evaluated ParFlow's performance and ability to simulate overland flow and integrated groundwater-surface water exchange. In my opinion, the manuscript is well-structured and written precisely. However, the authors should answer the following questions before the acceptance of the manuscript:

·         Incorporating spatially varying subsurface properties is critical to achieving accurate and reliable hydrologic simulations in the ParFlow model. However, the authors include simplified datasets and the absence of spatial heterogeneity, which may lead to biased simulation outputs. Please Explain.

·         Even though spatial heterogeneity was not considered in the study, the Root Mean Square Deviation (RMSD) between simulated and in-situ soil moisture values in the BJH case was very low, with values of 0.008 and 0.007 at 2 cm and 4 cm depths, respectively (Figure 13). The observation indicated that the ParFlow model could accurately simulate the hydrological response even with a spatially uniform and constant subsurface property assumption. Please elaborate.

Author Response

We thank Reviewer 2 for his/her thorough review, positive feedbacks, and constructive suggestions. All the comments are carefully addressed, and point-by-point responses are listed in the attachment. All responses can be found in the Word file.

Round 2

Reviewer 1 Report

No comments.